# Time-Limited Therapy with Belatacept in Kidney Transplant Recipients

**DOI:** 10.3390/jcm11113229

**Published:** 2022-06-06

**Authors:** Thibault Letellier, Delphine Kervella, Abderrahmane Sadek, Christophe Masset, Claire Garandeau, Cynthia Fourgeux, Victor Gourain, Jeremie Poschmann, Gilles Blancho, Simon Ville

**Affiliations:** 1Institut de Transplantation Urologie Néphrologie (ITUN), Centre Hospitalo-Universitaire Nantes, 44000 Nantes, France; thibault.letellier@chu-nantes.fr (T.L.); delphine.kervella@chu-nantes.fr (D.K.); christophe.masset@chu-nantes.fr (C.M.); claire.garandeau@chu-nantes.fr (C.G.); gilles.blancho@chu-nantes.fr (G.B.); 2CR2TI-U1064 The Center for Research in Transplantation and Translational Immunology, Université de Nantes, 44000 Nantes, France; cynthia.fourgeux@univ-nantes.fr (C.F.); victor.gourain@univ-nantes.fr (V.G.); jeremie.poschmann@univ-nantes.fr (J.P.); 3Biotechnology and Bio Resources Development Laboratory, Faculty of Sciences, Moulay Ismail University, Meknes 50050, Morocco; sadeksampi@gmail.com

**Keywords:** kidney transplantation, belatacept, calcineurin inhibitor, transcriptome, RNAseq

## Abstract

Introduction: In kidney transplant recipients, belatacept is usually pursued indefinitely after it has been started. In the setting of the belatacept shortage and after having evaluated the benefit–risk ratio, we established a strategy consisting of time-limited belatacept therapy/transient calcineurin inhibitor withdrawal, whose results are analyzed in that study. Methods: We considered all the kidney transplant recipients that had been switched from conventional immunosuppressive therapy to belatacept and then for whom belatacept has been withdrawn intentionally. Furthermore, in the first 8 patients, we assessed changes in peripheral blood mononuclear cells (PBMC) transcriptome using RNAseq before and 3 months after belatacept withdrawal. Results: Over the study period, 28 out of 94 patients had belatacept intentionally withdrawn including 25 (89%) switched to low-dose CNI. One rejection due to poor compliance occurred. The eGFR after 12 months remained stable from 48 ± 19 mL.1.73 m^−2^ to 46 ± 17 mL.1.73 m^−2^ (*p* = 0.68). However, patients that resumed belatacept/withdrew CNIs (*n* = 10) had a trend towards a better eGFR comparing with the others (*n* = 15): 54 ± 20 mL.1.73 m^−2^ vs. eGFR 43 ± 16 mL.1.73 m^−2^, respectively (*p* = 0.15). The only factor associated with belatacept resumption was when the withdrawal took place during the COVID-19 outbreak. Transcriptome analysis of PBMCs, did not support rebound in alloimmune response. Conclusions: These findings underpin the use of belatacept as part of a time-limited therapy, in selected kidney transplant recipients, possibly as an approach to allow efficient vaccination against SARS-CoV-2.

## 1. Introduction

During the last decades, improvements in long-term graft survival have been slight [1]. From studies in heart transplant recipients [2] and a series of kidney transplant surveillance biopsies [3], it has been assumed that chronic allograft dysfunction was caused by calcineurin inhibitor (CNI) nephrotoxicity, and attempts to withdraw or minimize their use have been pushed. However, belatacept [4] first used de novo in association with Mycophenolate Mofetil (MMF) as part of a CNI-free regimen [5,6], has not become the gold standard due to an unexpectedly high rate of early acute rejection. Alternatively, belatacept has been used successfully as rescue therapy to allow CNI withdrawal in patients with poor renal function in the early months post-transplantation or undergoing severe CNI-related adverse events [7,8,9,10,11,12]. Currently, belatacept is continued indefinitely, in line with the persistent fear for CNI nephrotoxicity. However, whilst there is no doubt about the acute and reversible vasoconstrictive effects of CNI on the glomerular tuft, their long-term harmfulness on the kidney has been challenged. The specificity of chronic histological lesions induced by calcineurin inhibitors has been questioned [13] and a large fraction of long-term kidney transplant failures are now attributed to antibody-mediated rejection [14,15,16]. Conversion from tacrolimus to belatacept is undoubtedly a suitable option in patients with poor kidney function in the few months posttransplantation, a time when a high trough level of tacrolimus is required. Farther in the post-transplantation period, belatacept’s advantage is less obvious, as lower trough levels of CNI, unlikely to be nephrotoxic, are sufficient to prevent rejection. Based on that hypothesis, we decided to stop belatacept and resume CNI at a low dose in kidney transplant recipients (KT) previously converted to belatacept. This monocentric retrospective study assessed the feasibility of that strategy by reporting its outcomes. Furthermore, for the first eight patients, we evaluated the changes in peripheral blood mononuclear cells (PBMC) transcriptome before the belatacept withdrawal and 3 months after, using RNAseq.

## 2. Methods

### 2.1. Study Population

We carried out a monocentric study on all consecutive adult patients who underwent kidney transplantation between 1 January 2010, and 31 December 2019, at the Nantes University Hospital. Firstly, from all patients that have been switched to belatacept, we considered those for whom belatacept has been stopped during follow-up. Meaning that those deceased, returned in dialysis, with missing data and who remained on belatacept at the last follow-up we excluded. Secondly, we differentiated those for whom the decision to stop belatacept had been planned with the clear intention of time-limited belatacept therapy (see below population section of the results) and those for which the discontinuation was driven by other causes and excluded the latter ones.

All data were extracted from the French, multicentric, observational and prospective DIVAT cohort of transplanted patients (Données Informatisées et VAlidées en Transplantation; www.divat.fr, CNIL final agreement, decision DR-2025-087 [No914184] 15 February 2015).

### 2.2. Immunosuppression

Before 2016, all low-immunological risk patients received induction immunosuppression consisting of 20 mg of basiliximab on day 0 and day 4 (Simulect, Novartis, Basel, Switzerland) and 250 mg bolus of methylprednisolone. The standard post-transplant immunosuppression includes calcineurin inhibitor (CNI), namely tacrolimus (trough level between 6 and 10 ng/dL) or cyclosporine (CSA; trough level between 125 and 200 ng/mL) and mycophenolate mofetil (MMF; 500–1000 mg/BID) or mycophenolic acid (MPA; 360–720 mg/BID).

High-immunological risk patients (definite by panel-reactive antibody (PRA) > 75%) received induction immunosuppression with rabbit antithymocyte globulin (rATG; Thymoglobulin, Genzyme, Cambridge, MA, USA) 6 mg/kg and a 250 mg bolus of methylprednisolone followed by triple immunosuppression including CNI, MMF or MPA, and prednisone.

Our standard protocol planned steroids withdrawal between 1 and 3 months, but some patients remained on triple therapy (rejection and/or high-immunological risk patients) or dual therapy with CNI and steroids in case of MMF/MPA withdrawal due to poor clinical tolerance and/or infections.

After 2016, low-immunological risk patients received a low dose of rATG (3 mg/kg) as induction therapy instead of basiliximab.

Conversion from conventional therapy to belatacept in some patients (Nulojix, Bristol-Myers Squibb, New York, NY, USA) was decided on an individual level at the discretion of clinicians, mainly in patients with poor renal function and suspected to have adverse events related to tacrolimus or cyclosporine exposure. Belatacept administration schedule was 5 mg/kg, repeated at 2 and 4 weeks, then every 4 weeks. In most cases, tacrolimus was tapered after belatacept introduction with the posology halved after 2 weeks followed by a complete withdrawal after one month.

At belatacept discontinuation, other immunosuppressive drugs such as tacrolimus or sirolimus were resumed on the day of the last belatacept injection.

### 2.3. Available Data

Recipient characteristics collected were gender, age, number of previous transplants, initial renal disease, and renal replacement therapy, history of hypertension or diabetes, presence of anti- Human Leukocyte Antigen (HLA) antibody, and DSA before transplantation.

Donor features were age, donor type (living, brain-dead, or non-heart-beating). Baseline transplantation parameters included cold ischemia time, number of HLA A-B-DR incompatibilities, induction therapy, initial maintenance treatment, use of steroids and delayed graft function. Estimated Glomerular filtration rate (eGFR) was estimated using the Modification of Diet in Renal Disease (MDRD) Study equation [17]. During post-transplantation follow-up, in accordance with guidelines regarding outpatient surveillance of kidney transplant recipients, frequent clinical and biological assessments were conducted.

Data collection stopped upon last known visit, return to dialysis, or death.

### 2.4. Statistical Analysis

The estimated Glomerular Filtration Rate (eGFR) values were calculated using the Modification of Diet in Renal Disease (MDRD) equation and reported as mean (SD). Qualitative variables were analyzed using Chi-square or Fisher’s exact test. Student’s *t*-test was used to compare quantitative variables. After having checked for normality assumption, one-way repeated measures ANOVA was used to analyze eGFR kinetic, pairwise paired *t*-tests were used for comparison between two time-point. Analysis was conducted with RStudio version 1.4.1106 (Joseph J. Alaire, Boston, MA, USA).

### 2.5. Analysis of the Transcriptome Changes of PBMC, before and 3 Months after the Belatacept Discontinuation Using RNAseq

For the 8 first patients for whom we had planned belatacept discontinuation in the context of the belatacept shortage (see below Section 3), residual blood samples were kept for scientific interest at 4 different time points: the day of belatacept discontinuation (D0) and after 1 month (M1), 2 months (M2) and 3 months (M3). The subject’s written consent was collected, and the samples stored were integrated into the collection of human biological samples DIVAT (n° DC-2011-1399 at the Ministry of Research and having obtained a favorable decision from the CPP Ouest IV on 7 April 2015).

PBMC were isolated by gradient protocol (Ficoll^®^). Total RNA was extracted from all samples with the TRIzol^®^ isolation protocol followed by QIAGEN RNeasy Micro clean-upprocedure. RNA samples with a > 7 RIN score were used. For 3′ DGE profiling, RNA-sequencing protocol was performed according to our implementation of Soumillon et al. protocol [18,19]. Briefly, the libraries were prepared from 10 ng of total RNA per sample (*n* = 26). The mRNA poly(A) tails were tagged with universal adapters, well-specific barcodes, and unique molecular identifiers during template-switching reverse transcription. Barcoded cDNAs were then pooled. 200 ng of cDNAs were amplified and fragmented using a transposon-fragmentation approach which enriches for 3′ ends of cDNAs (Nextera DNA Flex library prep ref 20,015,825 and 20,015,826 from Illumina). A library of 350–800 bp was run on a 100 cycles SP run on Novaseq6000 at GenoA IRS-un platform facility (Nantes). Samples were demultiplexed and aligned on the hg19 genome using the 3′ SRP pipeline.

The primary analysis of DGEseq data including, quality controls of reads, demultiplexing, read mapping, and quantification of gene expression, was carried out as described in Charpentier et al. [20] Normalization of gene expression and differential expression analysis were both performed with DESeq2 [21]. *p*-values were adjusted with the False Discovery Rate method and genes with an adjusted *p*-value less than 0.05 were considered as differentially expressed (DEG). Conditions for the comparison corresponded to the D0 and M3. Gene ontology enrichment analyses were performed using Enrich [22]. The same analysis pipeline was implemented with the dataset from peripheral blood obtained from KT recipients at the time of their 3 months protocolar biopsy [23]. Raw RNAseq was obtained from The European Nucleotide Archive (ENA) database (https://www.ebi.ac.uk/ena (accessed on 1 June 2021)), understudy accession PRJNA492956.

## 3. Results

### 3.1. Population

In the context of the belatacept shortage, a strategy of time-limited belatacept therapy was set out, that consisted of reconsidering its prescription in all patients that had been treated with belatacept for at least one year. Discontinuation and concomitant resumption of CNI or mTOR inhibitor at low dose was offered to patients that fulfilled the following criteria: (i) steady eGFR > 30 mL/min/1.73 m^2^, (ii) low immunological risk according to the judgment of their referring practitioner. In March 2020, the COVID-19 pandemic threatened to overwhelm the hospital’s capacities, and attendance at day-hospital (then legally required to administrate belatacept) was considered unsafe, the implementation of the process was sped up.

During the study period, 94 patients were switched to belatacept, then it was discontinued in 36 patients, including 28 for whom the decision to stop belatacept has been planned intentionally as outlined above (Figure 1). In 10 of them, the decision had been made at the time of the COVID-19 outbreak.

In total, 12 out of 28 (42%) were male, their mean age was 51 years, and most of them (*n* = 25, 89%) were transplanted for the first time, mostly from a brain-dead donor (*n* = 21, 75%) including 13 (62%) extended-criteria donor. Whereas induction therapy was predominantly antithymocyte globulin (*n* = 16, 57%), initial maintenance therapy was almost always an association of tacrolimus (*n* = 27, 96%), mycophenolate derivatives (*n* = 28, 100%), and prednisone (*n* = 27, 96%). Only 1 patient endured an acute rejection before belatacept introduction (active ABMR). Incidence of delayed graft function was 36% (*n*= 10).

Initial conversion to belatacept had been justified by poor renal function (*n* = 21, 75%) or suspected CNI-related adverse events (*n* = 6, 21%). In one case, the reason for the switch was an active ABMR. The switch timing was equivalently distributed between early (<3 months, *n* = 9, 32%), late (>1 year, *n* = 11, 39%) and intermediate (*n* = 8, 29%) posttransplant. Except one, all received other immunosuppressive drugs: mycophenolate derivative (*n* = 15, 54%), steroids (*n* = 4, 14%) or both (*n* = 8, 29%), only two patients continued a low dose CNI therapy either with tacrolimus or cyclosporin, for focal and segmental glomerulosclerosis relapse and ABMR before conversion, respectively (Table 1).

As illustrated in Figure 2, the mean eGFR before conversion to belatacept was 33 ± 17 mL.1.73 m^−2^. It improved swiftly after 3 months (43 ± 20 mL.1.73 m^−2^, *p* < 0.001) then steadily to reach 46 ± 12 mL.1.73 m^−2^ at 1-year post-conversion (*p* = 0.003).

### 3.2. Belatacept Discontinuation

Belatacept discontinuation occurred at a mean time post-transplant of 41.2 (12.9–188.4) months when patients had been given belatacept for 21.6 (4.3–50.8) months. Except for three cases (two monotherapies with mycophenolate derivative and one with mTOR inhibitor), the new immunosuppressive regimen was based on tacrolimus (association with mycophenolate derivative *n* = 17, with azathioprine *n* = 2, with mTOR inhibitor *n* = 2, alone *n* = 2).

As illustrated in Appendix A, tacrolimus trough level was maintained low, between 4 and 6 ng/mL.

After belatacept discontinuation, only one patient experienced an acute rejection due to poor adherence to treatment that quickly led to graft loss (severe mixed rejection). One patient (83 years old) died of an unknown cause at his residence, and one was lost to follow-up because of relocation, both shortly after belatacept discontinuation.

Belatacept was resumed in 10 out of the remaining 25 patients after a mean time of 5.1 (1–15.3) months (Figure 3A). Mentioned reasons were adverse events suspected to be tacrolimus-related (headache (*n* = 2), digestive disorders (*n* = 2), diabetes (*n* = 1)), patient request (*n* = 3), and finally decline of the renal function (*n* = 2). Table 2 presents patients’ characteristics according to their status regarding belatacept resumption or not.

Remarkably, 8 out of the 10 patients that resumed belatacept had been discontinued at the time of the COVID-19 pandemic (*p* < 0.01); among them were all patients who had belatacept reintroduced at their request. No other parameter was associated with belatacept resumption including the eGFR at the time of the belatacept withdrawal or the type of immunosuppressive drugs that have been resumed.

Three months after belatacept discontinuation and simultaneous resumption of other immunosuppressive drugs, a slight drop in eGFR was observed, from 48 ± 19 mL.1.73 m^−2^ to 45 ± 15 mL.1.73 m^−2^ (*p* = 0.12). Afterward, the mean eGFR remained stable: 46 ± 17 mL.1.73 m^−2^ at 1-year post-discontinuation (*p* = 0.20) (Figure 3B). When we differentiated patients regarding their status after the belatacept discontinuation, i.e., whether they had resumed belatacept or not, we did not observe any significant change in eGFR (Figure 3C). One-year post-discontinuation, there was a better eGFR in those who resumed belatacept and withdrew the CNIs vs. the others: 54 ± 20 mL.1.73 m^−2^ vs. eGFR 43 ± 16 mL.1.73 m^−2^, respectively, however non-significant (*p* = 0.15), in line with the reversible acute hemodynamic effect of the CNIs on the glomerular tuft. 

### 3.3. Adverse Events and Allo-Sensitization

CNI-free immunosuppressive regimen using belatacept not only has been demonstrated to improve renal function but also to avoid CNI-related adverse events and decrease the development of DSA. As described in Table 3, we did not identify a significant association between hypertension, the mean number of antihypertensive medications, diabetes, according to the periods of the study (at transplantation, before conversion from CNI to belatacept, under belatacept, and after belatacept discontinuation and resumption of CNI). As expected, there was a trend towards more infections in the immediate post-transplantation period (before conversion to belatacept), especially pyelonephritis.

As already mentioned, after belatacept discontinuation, one patient experienced a severe rejection due to poor adherence. Except for this patient none developed either rejection or DSA.

### 3.4. RNA-seq Analysis of PBMCs before and after Belatacept Discontinuation

In the first eights patients for whom belatacept was withdrawn on purpose, who all resumed with CNIs, we assessed by RNAseq changes in peripheral blood mononuclear cells (PBMC) transcriptome before (D0) and after that event (1 month (M1), 2 months (M2), and 3 months(M3)). As shown in Figure 4A, 69 genes were differentially expressed (DEGs) at M3 versus D0 (40 downregulated and 29 upregulated), versus only 14 and 16 at M1 and M2, respectively. Gene ontology enrichment analysis in the biological process revealed that upregulated genes were involved in the regulation of respiratory burst involved in the inflammatory response and the negative regulation of T cell receptor signaling pathway whereas downregulated genes were involved in co-translational protein targeting to the membrane and negative regulation of leukocyte degranulation (Appendix A). Remarkably, the product of one of the most negatively regulated gene, hematopoietic cell kinase (HCK), a member of the Src family of protein tyrosine kinases, has previously been identified as a key driver of fibrosis in transplant recipients [24]. Although how precisely it impacts the progression of fibrosis has not been elucidated, it is thought to involve both immune and non-immune cells. Indeed dasatinib, (an inhibitor of the Src kinase family skewed towards HCK) has been shown to decrease inflammation in different conditions including experimental allograft [25]. To go further we compare the DEGs between D0 and M3 identified in our patients, with those associated with subclinical rejection in a previously published RNAseq dataset performed on peripheral blood collected from 88 KT recipients at the time of 3 months surveillance biopsy [23]. Among the 10 genes common in both gene sets, those associated with the immunological process: HCK, FCGRT (Fc Fragment of IgG Receptor and transporter), and S100A9 were downregulated 3 months after the switch between belatacept and tacrolimus but upregulated in patients with a subclinical rejection on 3 months protocolar biopsies. Additionally, EZN, whose product ezrin is involved in the negative regulation of the TCR signaling pathway [26] was upregulated after 3 months in patients converted toward tacrolimus but downregulated in those with subclinical rejection (Figure 4B,C).

## 4. Discussion

Our study reports for the first time the implementation of a time-limited belatacept therapy or in other words a transient CNI withdrawal mostly in patients with poor renal function in the early period post-transplantation. After the conversion from belatacept to another immunosuppressive regimen containing low dose CNI in most cases, we observed a slight drop in eGFR without subsequent change at 1-year post-conversion and did not notice either acute rejection/sensitization or infectious/metabolic adverse event. Remarkably, in patients in whom belatacept was eventually resumed/CNI withdrawn, the eGFR rapidly improved, arguing for a hemodynamic and reversible effect of CNI on the glomerular tuft. Finally, in a group of patients, changes in RNA-seq transcriptome profiles of PBMCs did not suggest a rebound of the alloimmune response after belatacept withdrawal.

Nowadays, belatacept is increasingly used as part of a conversion strategy with the purpose to withdraw CNI in the increasing patient population with poor renal function a few months after transplantation. Studies reporting the outcomes of this strategy have constantly found an improvement of renal function, certainly caused by the relaxation of the glomerular tuft vasoconstriction in kidneys previously injured by the transplantation [7,8,9,10,11,12]. Up to now, whether CNI avoidance should be indefinitely continued once kidney function improved had never been addressed.

Only one study has reported a multicentric series of 44 patients converted from belatacept to another immunosuppressive drugs [27]. Results showed a significant decrease in eGFR from 44 to 36 mL/min/1.73 m^2^ and the authors concluded that belatacept should not be stopped. However, in many cases, belatacept withdrawal occurred in the setting of acute complications, sometimes serious, making the interpretation of the results difficult. When the authors focused on the 13 patients for whom belatacept had been withdrawn apart from any complication, no significant change was observed in eGFR. Consistent with ours, this finding suggests that the conversion from belatacept to tacrolimus might be safe when accomplished on purpose, in patients with a suitable kidney function and with careful monitoring of the tacrolimus trough levels that should be maintained low around 6 ng/mL, enough when the immunological risk is low [28].

Beyond a drop in GFR, the other theoretical risk of changing immunosuppressive drugs is to favor acute rejection/allo-sensitization. We did not observe such an event except in one case, related to non-compliance. This case highlights that when belatacept is introduced to ensure efficient immunosuppression, as it has been proposed especially in young patients [29], it should not be stopped before a thorough assessment of the patient’s treatment adherence. Additionally, the changes in peripheral blood transcriptome profiles between D0 and M3 did not argue for a rebound of the alloimmune response, when examined separately as well as when compared with the DEGs known to be associated with subclinical rejection [23]. However, a drawback of bulk transcriptomic is that it does not allow to identify which cell type explains the difference in gene expression observed between two conditions, especially regarding immune cells in whom many genes are expressed by a large range of cell subsets. Similarly, GO enrichment analysis can be flawed as the same pathway might have an opposite functional impact according to the cell type involved. For instance, although the TCR signaling is thought to be pro-inflammatory it may be suppressive if the cells involved are regulatory T cells. Only the use of methods allowing to reach a single-cell scale, such as flow cytometry with a limited number of markers though, or single-cell RNAseq, could overcome these limitations and deciphered accurately what occurs after withdrawing belatacept and resuming the CNIs.

It is worth noting that 10 out of 25 patients resumed belatacept, which might be interpreted as a failure of our strategy. However, only 3 patients had it reintroduced for objective reasons, such as kidney function degradation (*n* = 2) and diabetes triggered by CNI (*n* = 1), and, importantly, all returned after the CNI withdrawal. In the others, causes were adverse events supposed to be CNI-related (*n* = 4) and at the patient’s request (*n* = 3).

Only the timing of the belatacept discontinuation, namely when it had been implemented at the time of the COVID-19 outbreak, was significantly associated with belatacept resumption. We may assume that in this very specific setting, patients felt compelled to stop belatacept, explaining the high rate of resumption afterward. That highlights the importance of therapeutic patient education to ensure adherence to treatment change [30].

Our study has some obvious limitations, mainly the low number of patients and the limited follow-up after the conversion from belatacept to other immunosuppressant drugs. Consequently, we cannot ensure that long-term kidney function could not have been impacted by CNI exposure. However, chronic CNI nephrotoxicity is increasingly questioned. In this regard, our finding that the expression of HCK, a gene found to be a key driver on kidney fibrosis [24], was significantly decreased after the conversion from belatacept to tacrolimus, may be cautiously interpreted as reassuring.

Our findings raise the question of the potential benefit of using belatacept in a time-limited way, apart from the avoidance of iv injection and a potential medico-economic advantage. Except for the well-known risk of EBV-induced lymphoproliferative disease in seronegative recipients, and based on the data from BENEFIT [5,6] along with some in vitro data, belatacept has been deemed as safe regarding the risk of serious opportunistic infection. However, no specific evaluation had been conducted in the very different setting of conversion from CNI to belatacept in patients often older, with a poor eGFR. Recently, concerns have been raised by some reports [31,32] and the result of a multicentric cohort of 280 KT recipients switched to belatacept, in which as many as 42 opportunistic infections were reported [33]. In a retrospective matched study analyzing CMV disease characteristics after belatacept conversion [34], authors found a sharply increased risk of CMV disease (17.7% vs. 2.8%) associated with older age and low eGFR at conversion. Remarkably, the pattern of CMV disease was unusual: occurring in seropositive patients, severe and surprisingly late (>1-year-post-conversion) in most cases. Finally, evaluation of mRNA SARS-CoV-2 vaccine in kidney transplanted recipients have demonstrated the worrying effectiveness of belatacept in inhibiting immune response to vaccination when compared with CNI. Although the rate of seroconversion was between 30 and 50% in patients treated with a CNI-based regimen, it plunged to 0 to 6% in patients receiving belatacept [35,36,37]. Moreover, the administration of a third vaccine dose, that has been demonstrated to improve the immunogenicity of the mRNA vaccine in KT recipients remains dramatically inefficient in patients treated with belatacept [38]. The accumulating evidence that tends to demonstrate that prolonged use of belatacept is associated with an increased risk of infection, especially in older patients with poor renal function at conversion, could justify a strategy of time-limited therapy with belatacept, once the improvement in kidney function is firmly established. Depending on the evolution of the COVID-19 pandemic, one could also consider stopping belatacept to vaccinate patients. In this respect, our data supporting that the reintroduction of CNI at a low dose is safe are helpful.

In all, our findings underpinned the concept of a time-limited belatacept therapy in a selected group of kidney transplant recipients. Further evaluation including well-conducted prospective studies with a long-term follow-up investigating the benefit–risk ratio of that strategy are needed.

## Figures and Tables

**Figure 1 jcm-11-03229-f001:**
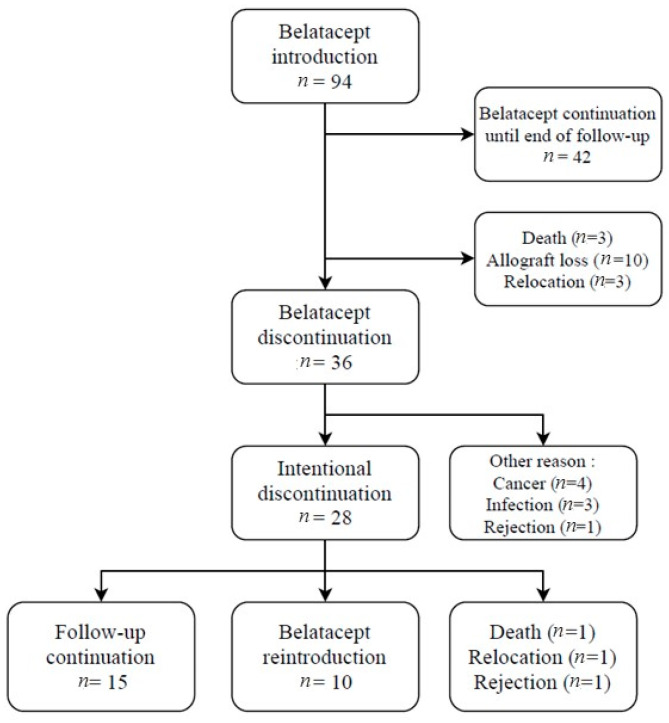
Flowchart of the study.

**Figure 2 jcm-11-03229-f002:**
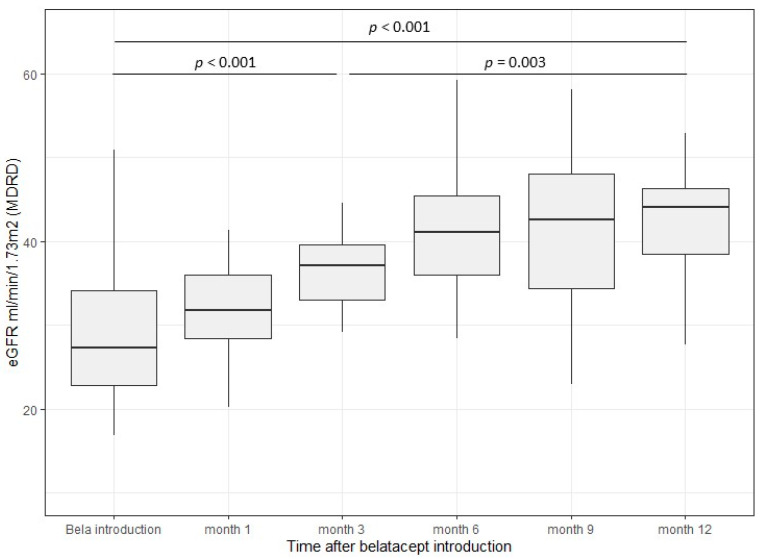
Effect of belatacept introduction/CNIs withdrawal on eGFR (*n* = 28).

**Figure 3 jcm-11-03229-f003:**
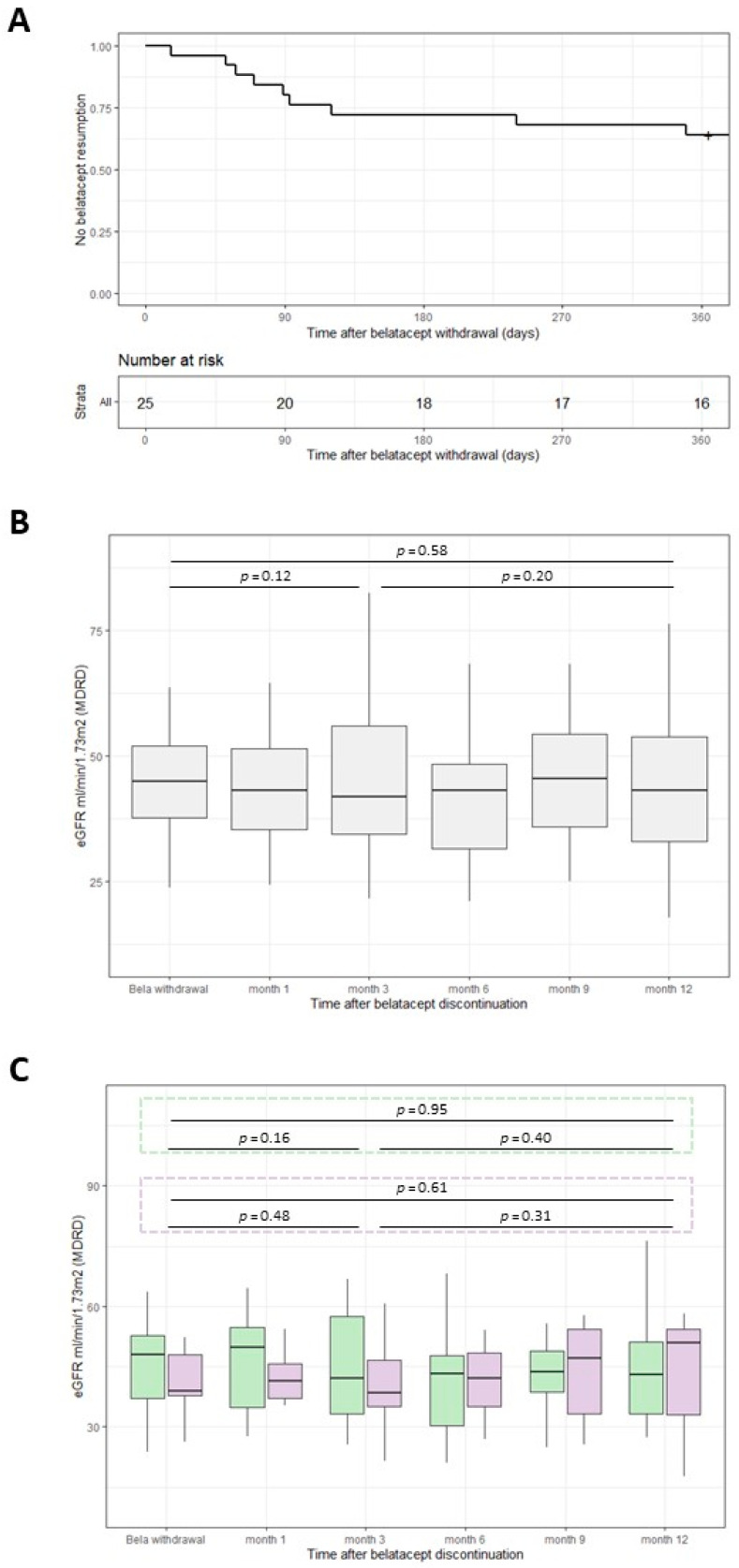
Effect of belatacept discontinuation/CNIs resumption on eGFR (*n* = 25). (**A**) Kaplan–Meier analysis of time to belatacept resumption during the first year post-belatacept-discontinuation; (**B**) evolution of the eGFR during the first year after belatacept discontinuation (*n* = 25); (**C**) evolution of the eGFR during the first year after belatacept discontinuation according to their belatacept-status: green boxplots correspond to patients without belatacept (*n* = 15) and the purple ones to patients having resumed belatacept as shown in A (*n* = 10).

**Figure 4 jcm-11-03229-f004:**
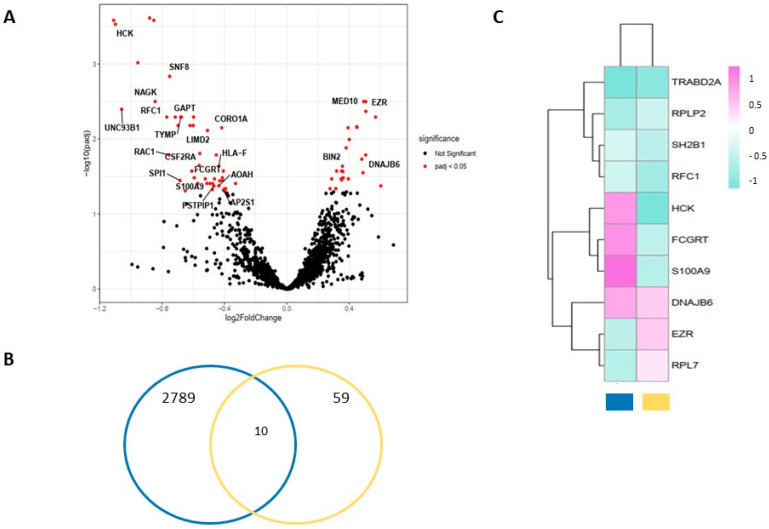
RNA-sequencing profiles analysis of peripheral blood mononuclear cells (PBMC) collected before (D0) and 3 months (M3) after belatacept withdrawal. (**A**) Volcano plot representing the Differential Expressed Genes (downregulated as negative fold change and upregulated as positive fold change) between D0 and M3; (**B**) Venn diagram displaying the relationship of DEGs between D0 and M3 in our dataset (yellow circle) and those associated with subclinical rejection in a previously published RNAseq dataset performed on whole blood collected from 88 KT recipients at the time of 3-month surveillance biopsy [24] (blue circle); (**C**) Heatmap displaying the regulation of the 10 DEGs that overlapped in both datasets. Blue square corresponds to DEGs between D0 and M3, and yellow square to DEGs between subclinical rejection or not on 3 months protocolar biopsy. The positive value/violet color scale represents upregulated genes, whereas the negative value/cyan color scale represents downregulated genes.

**Table 1 jcm-11-03229-t001:** Patient’s characteristics at baseline and belatacept conversion. (*n* = 28 patients for whom belatacept has been withdrawn intentionally).

Baseline Features		
**Sex (%)**	Female	16 (57.1)
	Male	12 (42.9)
**Age (mean (SD))**		50.75 (17.19)
**Initial nephropathy (%)**	Other	6 (21.4)
	Glomerulopathy	7 (25.0)
	Tubulo-interstitial or uropathy	10 (35.7)
	Vascular	3 (10.7)
	ADPKD	2 (7.1)
**Extrarenal epuration therapy (%)**	Preemptive	5 (17.9)
	Peritoneal dialysis	5 (17.9)
	Hemodialysis	18 (64.3)
**Diabetes (%)**	No	27 (96.4)
	Yes	1 (3.6)
**Donor type (%)**	Brain death	21 (75.0)
	Circulatory death	3 (10.7)
	Living	4 (14.3)
**CMV status (%)**	R+	11 (39.3)
	D−/R−	16 (57.1)
	D+/R−	1 (3.6)
**Transplantation rank (%)**	1	25 (89.3)
	2	2 (7.1)
	3	1 (3.6)
**DSA (%)**	No	22 (78.6)
	Yes	2 (7.1)
	NA	4 (14.3)
**HLA incompatibilities (median [IQR])**		4.00 [3, 5]
**Induction treatment (%)**	Basiliximab	12 (42.9)
	Antithymocyte globulin	16 (57.1)
**Cyclosporin (%)**	No	27 (96.4)
	Yes	1 (3.6)
**Tacrolimus (%)**	No	1 (3.6)
	Yes	27 (96.4)
**Mycophenolate derivative (%)**	Yes	28 (100.0)
**Corticosteroids (%)**	No	1 (3.6)
	Yes	27 (96.4)
**Delayed graft function (%)**	No	18 (64.3)
	Yes	10 (35.7)
**Features at belatacept conversion**		
**DSA (%)**	No	22 (78.6)
	Yes	5 (17.9)
	NA	1 (3.6)
**Diabetes (%)**	No	25 (89.3)
	only dietary rules	2 (7.1)
	Insulin	1 (3.6)
**Hypertension (%)**	No	8 (28.6)
	Yes	20 (71.4)
**Rejection (%)**	No	27 (96.4)
	Yes	1 (3.6)
**Cause of switch (%)**	Active ABMR	1 (3.6)
	Poor CNI tolerance	6 (21.4)
	Impaired function	21 (75.0)
**Time since transplantation (months, mean (SD))**	19.5 (37.6)	
**Time since transplantation (%)**	Early (<3 months)	9 (32.1)
	Intermediate (3 to 12 months)	8 (28.6)
	Late (>12 months)	11 (39.3)
**Last eGFR before switch (mL/min/1.73 m^2^) mean (SD)**		33 (17)
**Immunosuppressive drug associated with belatacept**		
**Tacrolimus (%)**	No	26 (92.9)
	Yes	2 (7.1)
**Cyclosporin (%)**	No	27 (96.4)
	Yes	1 (3.6)
**Other immunosuppressants (%)**	No	1 (3.6)
	Corticotherapy	4 (14.3)
	MMF/MPA	15 (53.6)
	Corticotherapy + MMF/MPA	8 (28.6)
**Rejection (%)**	No	27 (96.4)
	Yes	1 (3.6)

Abbreviations. SD: standard deviation; ADPKD: autosomal dominant polycystic kidney disease; DSA: donor-specific antibody; IQR: inter-quartile range, ABMR: antibody-mediated rejection; CNI: Calcineurin inhibitor; MMF: mycophenolate mofetil; MPA: mycophenolic acid.

**Table 2 jcm-11-03229-t002:** Patient’s characteristics according to their belatacept resumption status.

Characteristics		No Belatacept Resumption *n* = 15	Belatacept Resumption *n* = 10	*p*
**Age (mean (SD))**		51.4 (17.3)	49.9 (15.2)	0.826
**Sex (%)**	Female	8 (53.3)	8 (80)	0.349
	Male	7 (46.7)	2 (20)	
**Transplantation rank (%)**	1	13 (86.7)	9 (90)	0.242
	2	2 (13.3)	0 (0)	
	3	0 (0)	1 (10)	
**Delayed graft function (%)**	No	10 (66.7)	6 (60)	1.000
	Yes	5 (33.3)	4 (40)	
**Belatacept duration (mean (SD))**		578.4 (327.6)	823.5 (341.3)	0.085
**Cause of switch (%)**	Active ABMR	1 (6.7)	0 (0)	0.629
	Poor CNI tolerance	4 (26.7)	2 (20)	
	Poor function	10 (66.7)	8 (80)	
**Cause of discontinuation (%)**	Standard protocol	13 (86.6)	2 (20)	**<0.01**
	COVID-19 pandemic	2 (13.4)	8 (80)	
**Last eGFR under belatacept (mL/min/1.73 m^2^) (mean (SD))**		50 (22)	44 (13)	0.534
**Corticosteroids (%)**	No	11 (73.3)	6 (60)	0.793
	Yes	4 (26.7)	4 (40)	
**mTOR-inhibitor (%)**	No	11 (73.3)	9 (90)	0.610
	Yes	4 (26.7)	1 (10)	
**CNI (%)**	No	2 (13.3)	1 (10)	1.000
	Yes	13 (86.7)	9 (90)	
**CNI + Mycophenolate derivative (%)**	No	5 (33.3)	6 (60)	0.366
	Yes	10 (66.7)	4 (40)	
**CNI + mTOR inhibitor (%)**	No	12 (80.0)	9 (90)	0.911
	Yes	3 (20.0)	1 (10)	

Abbreviation: SD: standard deviation; ABMR: antibody-mediated rejection; CNI: calcineurin inhibitor; COVID: coronavirus disease; mTOR: mammalian target of rapamycin.

**Table 3 jcm-11-03229-t003:** Side effects according to the belatacept exposure status.

		At Transplantation *n* = 28	Before Belatacept ^a^ *n* = 28	Under Belatacept ^b^ *n* = 28	After Belatacept ^c^ *n* = 19 ^d^	*p*
**Diabetes**						0.60
	No (%)	27 (96.4)	25 (89.3)	25 (89.3)	17 (89.4)	
	LDI only (%)	0	2 (7.1)	2 (7.1)	1 (5.3)	
	Oral therapy (%)	0	0	0	1 (5.3)	
	Insulin (%)	1 (3.6)	1 (3.6)	1 (3.6)	0	
**Hypertension**						0.71
	No (%)	10 (35.7)	9 (32.1)	10 (35.7)	4 (21.1)	
	Yes (%)	18 (64.3)	19 (67.9)	18 (64.3)	15 (78.9)	
**Mean number of anti-hypertensive medication**						0.35
		1.21	1.18	1.14	1.21	
**DSA**						0.65
	No (%)	25 (89.3)	23 (82)	24 (85.6)	15 (78.9)	
	Yes (%)	2 (7.1)	5 (18)	3 (10.7)	3 (15.9)	
	NA (%)	1 (3.6)	0	1 (3.6)	1 (5.3)	
**Infections**						0.09
	Pyelonephritis (%)		9 (32.1)	3 (10.7)	1 (5.3)	
	Other bacterial (%)		3 (10.7)	3 (10.7)	2 (10.6)	
	Flu (%)		0	2 (7.1)	1 (5.3)	
	COVID-19 (%)		0	0	2 (10.6)	
	Other viral (%)		1 (3.6)	2 (7.1)	1 (5.3)	

a. Before belatacept: at last medical consultation before belatacept’s introduction. b. Under belatacept: at 12 months after its introduction. c. After belatacept: at 6 months after its discontinuation. d. *n* = 19: one patient died, one patient moved away and 7 resumed belatacept before 6 months. Abbreviations: LDI: lifestyle and dietary interventions; DSA: donor-specific antibody; NA: not available; COVID: coronavirus disease.

## Data Availability

The data presented in this study are available on request from the corresponding author.

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
