# Peer review of "Time-Limited Therapy with Belatacept in Kidney Transplant Recipients"

_jcm, 2022, doi:10.3390/jcm11113229_

Round 1
Reviewer 1 Report
This is a monocentric retrospective study describing the outcomes of stopping Belatacept and resume calcineurin inhibitors at low dose in patients with kidney transplant. The authors focused their analyses into renal function, adverse events, and the transcriptome analysis of peripheral blood mononuclear cells from patients, before and after Belatacept discontinuation. In general, the group of patients is well-described and justified, and the molecular analyses were well conducted. I only have one concern regarding the quality of discussion related the transcriptome findings.
Major:
The discussion regarding the transcriptome analysis is poor. Since PBMC are a mixture of cell populations (mainly monocytes and lymphocytes), How authors interpret the significant changes of gene expression in PBMC after belatacept withdrawal? In tubular cells, HCK may play a critical role in renal fibrosis, but there is no mention to its role in PBMC. Similarly, they mentioned that some up-regulated genes are involved on the regulation of TCR pathways. However, regulatory T cells require TCR signaling for their suppressive function, while other T-cells require TCR signaling for their pro-inflammatory effects. There is no major explanation about those results or interpretations in their findings relating the phenotype associated in patients.
Minor:
Introduction.
- Please, define KT in line 53.
Results.
- In line 264, authors repeat the PBMc’s definition. It is already defined before. Also in legend of figure 4.
Author Response
Reviewer responses :
This is a monocentric retrospective study describing the outcomes of stopping Belatacept and resume calcineurin inhibitors at low dose in patients with kidney transplant. The authors focused their analyses into renal function, adverse events, and the transcriptome analysis of peripheral blood mononuclear cells from patients, before and after Belatacept discontinuation. In general, the group of patients is well-described and justified, and the molecular analyses were well conducted. I only have one concern regarding the quality of discussion related the transcriptome findings.
Major:
The discussion regarding the transcriptome analysis is poor. Since PBMC are a mixture of cell populations (mainly monocytes and lymphocytes), How authors interpret the significant changes of gene expression in PBMC after belatacept withdrawal? In tubular cells, HCK may play a critical role in renal fibrosis, but there is no mention to its role in PBMC. Similarly, they mentioned that some up-regulated genes are involved on the regulation of TCR pathways. However, regulatory T cells require TCR signaling for their suppressive function, while other T-cells require TCR signaling for their pro-inflammatory effects. There is no major explanation about those results or interpretations in their findings relating the phenotype associated in patients.
Thanks you for raising these interesting points.
You are perfectly right that bulk transcriptomic did not allow to identify which cell type explains the difference in gene expression observed between two conditions, especially regarding immune cells in whom many genes are expressed by a large range of cell types. To overcome that limit other methods allowing to reach a single-cell scale are required, such as flow cytometry with the drawback of having a limited number of markers, or single-cell RNAseq that could have been ideally used. As you mentioned is also true when it comes to analyzing the result of a GO enrichment analysis, with the example of the TCR signaling that can drive pro-inflammatory as well as suppressive pathways depending on the cell type. We have now developed these limitations in the discussion and insisted that other data especially with single-cell scale are required to go deeper in the deciphering of what happened after withdrawing belatacept and resuming the CNIs :
However, a drawback of bulk transcriptomic is that it does not allow to identify which cell type explains the difference in gene expression observed between two conditions, especially regarding immune cells in whom many genes are expressed by a large range of cell subsets. Similarly, GO enrichment analysis can be flawed as the same pathway might have an opposite functional impact according to the cell type involved. For instance, although the TCR signaling is thought to be pro-inflammatory it may be suppressive if the cells involved are regulatory T cells. Only the use of methods allowing to reach a single-cell scale, such as flow cytometry with a limited number of markers though, or single-cell RNAseq, could overcome these limitations and deciphered accurately what occurs after withdrawing belatacept and resuming the CNIs.
The role of HCK in the progression of fibrosis has been raised in a transcriptomic study comparing biopsies with or without renal injury (Wei, JASN, 2017). A deep understanding of how HCK impacts fibrosis was however complex because HCK is expressed both by the kidney epithelial cells and the immune cells. Because of indirect argument, chiefly the fact that dasatinib (an inhibitor or the src kinase family member skewed towards HCK) decreased inflammation in different conditions including experimental allograft (Khatri, JEM, 2013), authors assumed that the role of HCK in driving fibrosis likely takes place in both non-immune and immune cells. Thus, the fact that it appears to decrease in the blood after belatacept withdrawal/CNI resuming argues for potential better control of the alloimmune response. We now develop that idea in the manuscript:
Although how precisely it impacts the progression of fibrosis has not been elucidated, it is thought to involve both immune and non-immune cells. Indeed dasatinib, an inhibitor of the Src kinase family skewed towards HCK) has been shown to decrease inflammation in different conditions including experimental allograft25.
Minor:
Introduction.
Please, define KT in line 53.
We have changed accordingly.
Results.
In line 264, authors repeat the PBMc’s definition. It is already defined before. Also in legend of figure 4.
We have changed accordingly
Reviewer 2 Report
The authors investigated the effect of belatacept on kidney transplant recipients.
There are several critical drawbacks to this study.
First of all, the aim of this study is obscure. I cannot understand what the authors wanted to show in this study. Inconsistent standards for the withdrawal skewed this study. They said belatacept was stopped due to the COVID-19 pandemic, but this drug was also stopped by physicians’ decision. or depended on the criteria. (i) steady eGFR > 30 ml/min/1.73m2, (ii) low immunological risk ac-161 cording to the judgment of their referring practitioner.
Moreover, the observation period was not limited to the COVID-19 era.
Second, the number of patients who were collected data was too small. From only ten patients’ data, they said there was a trend for better eGFR, although the p-value was 0.15 (not significant level)
Third, they conducted multiple comparisons regarding eGFR change, however, they did not use suitable statistical analysis methods, such as repeated measure ANOVA.
Fourth, I do not think it necessary to show the RNA-sequencing profiles analysis of peripheral blood mononuclear cells, which made the manuscript more complicated.
Author Response
Reviewer responses:
The authors investigated the effect of belatacept on kidney transplant recipients.
There are several critical drawbacks to this study.
First of all, the aim of this study is obscure. I cannot understand what the authors wanted to show in this study. Inconsistent standards for the withdrawal skewed this study. They said belatacept was stopped due to the COVID-19 pandemic, but this drug was also stopped by physicians’ decision. or depended on the criteria. (i) steady eGFR > 30 ml/min/1.73m2, (ii) low immunological risk according to the judgment of their referring practitioner. Moreover, the observation period was not limited to the COVID-19 era.
Thanks for your remark that helped us to clarify our message, and hopefully will make the readers more comfortable. The aim of our study was to evaluate the feasibility of using belatacept rather than CNI during a time window that ranges from few months after the transplantation (before the risk of rejection with belatacept seems too high) to 1-year posttransplantation or more (when CNI low-dose is safe making belatacept “useless” or let say less useful, moreover as discussed in the discussion some recent data argue that prolonged exposition to belatacept could favor opportunistic infection). We have changed the end of the introduction to put it clearer, especially we removed the notion of the shortage and COVID-19 which made it confusing:
Based on that hypothesis and driven by the shortage of belatacept then the COVID-19 pandemic we decided to stop belatacept and resume CNI at a low dose in kidney transplant (KT) recipients previously converted to belatacept. This monocentric retrospective study assessed the feasibility of that strategy by reporting the outcomes.
Criteria that made us offer belatacept withdrawal to the patients were constant across the study period:
-belatacept started after the transplantation
-treatment with belatacept for at least one year.
- steady eGFR > 30 ml/min/1.73m2.
- low immunological risk according to the judgment of their referring practitioner.
The opinion of the physician was only about that last point if the others were fulfilled. The shortage of belatacept, as well as the covid-19 pandemic, were not what caused the withdrawal but contextual elements that we think important not to neglect because they may impact the outcome, as we have found out. We have added a sentence to make the process more understandable:
In March 2020, for COVID-19 pandemic threatened to overwhelm the hospital’s capacities, and attendance at day-hospital (then legally required to administrate belatacept) was considered unsafe, the implementation of the process was sped up (only for patients that fulfilled the criteria listed above).
Second, the number of patients who were collected data was too small. From only ten patients’ data, they said there was a trend for better eGFR, although the p-value was 0.15 (not significant level).
We perfectly agree that the number of patients involved is small. However, we still think that our data warrants to be shared with the community because they address a relevant clinical question: is that safe to resume CNI after a period with belatacept ? We agree that our data are too poor for a full reply as we have already stated in the discussion, but the point is that nowadays except for one study that we extensively mentioned in the discussion section, no publication has reported kidney transplants outcome in such case. As we outlined in the discussion in the other study different patterns of patients were mixed up: those for whom belatacept was stopped because of complications (infectious disease and cancer mainly), not suitable to address the question above, and those for whom belatacept was stopped intentionally just as in our patients. In these former, the outcomes seemed to be favorable, but their conclusions were impaired by the low number of patients (n= 13). So even if our number of patients is small it is around double what has been published before. Thus, we think that our data are original enough to be published. Of course, other studies with prospective design and ideally a control group will be required to confirm our preliminary results. We have already pointed it, out in the discussion section and in the conclusion:
Our study has some obvious limitations, mainly the low number of patients and the limited follow-up after the conversion from belatacept to other immunosuppressant drugs. Consequently, we cannot ensure that long-term kidney function could not have been impacted by CNI exposure. […] Further evaluation including well-conducted prospective studies with a long-term follow-up investigating the benefit-risk ratio of that strategy are needed.
You are right regarding that our interpretation of the eGFR evolution when patients are differentiated between having or not resumed belatacept is not correct. We have changed accordingly:
However, One-year post-discontinuation, there was a trend towards a better eGFR in those having resumed belatacept and withdrew the CNIs vs the others: 54 ± 20 ml.1.73m-2 vs eGFR 43 ± 16 ml.1.73m-2 respectively, however non-significant (p = 0.15), that could represent the reversible acute hemodynamic effect of the CNIs on the glomerular tuft.
Third, they conducted multiple comparisons regarding eGFR change, however, they did not use suitable statistical analysis methods, such as repeated measure ANOVA.
We had used the nonparametric Friedman which may also be appropriate for that purpose. But you are right that when the distribution is normal repeated measure ANOVA is more suitable. So after having checked for the normality assumption that was true, we ran the analysis using repeated measure ANOVA. It is worth noting that the interpretation of the results did not change as we still observed a significant change after the introduction of the belatacept but no change after its withdrawal/resumption of the CNIs whichever the conditions (Figure 3). We have also specified in the statistics section which test was used for paired comparison (pairwise paired t-tests) that was missing in our first manuscript :
After having checked for normality assumption, one-way repeated measures ANOVA was used to analyze eGFR kinetic, pairwise paired t-tests was used for comparison between two time-point.
Fourth, I do not think it necessary to show the RNA-sequencing profiles analysis of peripheral blood mononuclear cells, which made the manuscript more complicated.
As we stated before the aim of our study was to address the feasibility of withdrawing/resuming CNIs. A potential risk, as ever when immunosuppressive drugs are switched, is to trigger rejection. As we know alloimmune response does not always lead to severe rejection but more often a subclinical process, a way to monitor more closely our patients would have been to perform a protocolar biopsy for instance 3 months after the switch. However, that option did not appear ethically suitable, and we aimed at using RNA sequencing profiles of PBMC as a surrogate marker of the alloimmune response/subclinical rejection. Indeed, we compared the DEG (differentiated expressed genes) that we found comparing before and after the withdrawal/resumption of CNI with those that have been published in the blood of patients with subclinical rejection on their protocolar biopsy. Thus, we think earnestly that this approach, though with some limitations as outlined by reviewer 1, is valuable and is consistent with the actual trend of seeking surrogate markers besides classical clinical outcomes.
Reviewer 3 Report
Authors should be congratulated for the intriguing work. The manuscript is well written and easily readable, and the methodology is robust. Tables and figures are clear and well-presented. The finding of using belatacept as part of a time-limited therapy, in selected kidney transplant recipients, possibly as an approach to allow efficient vaccination against SARS-Cov-2 is very intriguing.
Reviewer 4 Report
Considering the following:
Considering the following: structuring, relevance, uniqueness, completeness of materials, methodological and theoretical foundations are at a high level. The information presented in the paper has a clear sequence and structuring is in line with the scientific style.
Round 2
Reviewer 2 Report
I understand that the authors tried to improve the manuscript, however, there are still critical drawbacks to this study. Basically, the study design was not well structured.
As far as I have searched literature that studied belatacept rescue therapy in kidney transplant recipients, one report included 79 patients (Transplant International 2016; 29: 1184–1195), and another paper included 280 patients (Nephrol Dial Transplant 2020; 35: 336–345). The latter elucidated that belatacept increased the risk of Opportunistic infections, such as viral infection.
In my opinion, the number of patients is too small to draw any significant results in this study.
I do not think this study could prove the safety of belatacept in patients who underwent kidney transplantation.